# BiomiMETRIC Assistance Tool: A Quantitative Performance Tool for Biomimetic Design

**DOI:** 10.3390/biomimetics4030049

**Published:** 2019-07-10

**Authors:** Philippe Terrier, Mathias Glaus, Emmanuel Raufflet

**Affiliations:** 1Laboratoire d’Ingénierie pour le Développement Durable, École de Technologie Supérieure, 1100 Notre-Dame Ouest, Montréal, QC H3C1K3, Canada; 2Station Expérimentale des Procédés Pilotes en Environnement, Département de Génie de la Construction, École de Technologie Supérieure, 1100 Notre-Dame Ouest, Montréal, QC H3C1K3, Canada; 3Département de Management, HEC Montréal, 3000 Chemin de la Côte-Sainte-Catherine, Montréal, QB H3T 2A7, Canada; 4Institute EDDEC, Environnement, Développement Durable et Économie Circulaire, Montréal, QC H3T 2A7, Canada

**Keywords:** biomimetic methodology, life’s principles, engineering design tool, life-cycle analysis, sustainable design

## Abstract

This article presents BiomiMETRIC, a quantitative performance tool for biomimetic design. This tool is developed as a complement to the standard ISO 18458 Biomimetics—terminology, concepts, and methodology to quantitatively evaluate the biomimetics performance of a design, a project, or a product. BiomiMETRIC is aimed to assist designers, architects, and engineers to facilitate the use of the biomimetic approach beyond the existing frameworks, and to provide an answer to the following question: How can a quantitative evaluation of biomimetic performance be carried out? The biomimetic quantitative performance tool provides a method of quantitative analysis by combining the biomimetic approach with the impact assessment methods used in life-cycle analysis. Biomimetic design is divided into eight steps. The seventh step deals with performance assessment, verifying that the concept developed is consistent with the 10 sustainable ecosystem principles proposed by the Biomimicry Institute. In the application of the biomimetic quantitative performance tool, stone wool and cork are compared as insulation materials used in biomimetic architecture projects to illustrate the relevance and added value of the tool. Although it is bio-based, cork has a lower biomimetic performance according to the indicators used by the biomimetic quantitative performance tool presented in this article.

## 1. Introduction

The idea of combining biomimetic design and life-cycle analysis (LCA) was previously suggested [1,2], but no tools or methods for evaluating biomimetics performance have been developed. LCA is often seen as a tool for measuring environmental impacts and unsustainability, whereas biomimetic and nature-inspired design is aimed to generate environmentally positive innovations to achieve sustainability [1].

Several frameworks have been proposed in the field of biomimicry. On one hand, two ISO standards are available: ISO 18458 [3], Biomimetics—Terminology, Concepts and Terminologies, which provides a framework for biomimetic terminology and a method for use in science, industry, and education; and ISO 18459, Biomimicry—Biomimetic Optimization, which specifies the functions and areas of application of biomimetic optimization methods to extend the life and reduce the mass of components. On the other hand, the Biomimicry Institute has proposed a biomimetic design sequence [4], recommending that during the design process, the Ask Nature database [5] be consulted in order to discover the “principles of Nature.” The main difficulties for neophytes in the field of biomimicry are first to know what to look for in the database and then to identify the keywords for searching the database in order to select a strategy. During the biomimetic design process, it is necessary to consider the function of the product in order to discover in the database, what nature does to achieve this function and what strategy “life” has put in place to achieve it. Despite the availability of standards such as ISO 18458 [3] and the exhaustive amount of information proposed by the Biomimicry Institute [6], the biomimetic design method is being developed, and the use of these frameworks is limited in fields such as engineering. This method is divided into eight steps, which are presented in Figure 1.

The implementation of this approach has weaknesses and limitations [7], mainly at Step 7 in the assessment of biomimetic performance. Several tools for discovering solutions exist in nature [8,9,10], and some are based on computer-aided biomimetics software developed for the integration of biological knowledge in engineering to facilitate the search for solutions [9,11,12]. More than 40 tools are available [9] to facilitate the use of the biomimetic approach in design, architecture, and engineering, which are focused mainly on the steps involved in identifying and transposing biological models to technological applications. However, the approach is often qualitative, and the majority of publications are focused on bioinspiration and the technological solutions that result from it. Very few articles have dealt with a biomimetic methodology that could be used in design or engineering [13] with the emphasis on quantifying biomimetic performance. There are no quantification tools available for assessing the degree of inclusion of biomimicry in the solutions developed [9].

In order to address the biomimetic design methodology, this article proposes the development of BiomiMETRIC, a tool for quantifying biomimetic performance based on the following premises: The “Life’s Principles” [14] are useful, but unfortunately, they are not quantified, and they are often difficult to implement in the context of an engineering project. The environmental impact assessment methods used in life-cycle assessment (LCA), which are available in databases such as Ecoinvent [15,16], allow for the quantification of environmental impacts and damage, such as climate change, resource degradation, human health impacts, and ecological footprints. By combining the quantitative impact assessment methods used in LCA with the principles and strategies of living organisms used in biomimetic design, we structure a quantitative assessment of the biomimetic performance of a technological solution, concept, or design. This methodological approach will be developed in the form of a tool that will facilitate the implementation of the biomimetic design method. BiomiMETRIC is aimed to help designers and engineers choose quantitative impact methods from LCA to evaluate the “Life’s Principles” and to facilitate the assessment of biomimetic performance.

## 2. Context and Terminology in Biomimetics

In this Section, we will present important concepts in biomimicry and in life-cycle analysis, which will allow the better understanding of the development of BiomiMETRIC assistance tool.

### 2.1. Functions and Strategies of Living Organisms

Functions and strategies are two fundamental terms in the biomimetic lexicon. The function [17] of a natural system is usually an adaptation that helps the system survive and thrive. The function is implemented by the strategies of a living system. A biological strategy is a characteristic, mechanism, or process that performs a function for a living organism. The function to be achieved is therefore at the heart of the biomimetic design process. Carrying out a biomimetic design requires following an approach where it is necessary to question the function that the technological solution must achieve. The designer must approach the problem to be solved without unconsciously favouring an a priori solution. At a later stage in the biomimetic design sequence, it has been recommended to analyse the conformity of the solution developed according to the “Life’s Principles” [14] that normally govern sustainable ecosystems. However, neither the approach proposed by the Biomimicry Institute nor the guidelines set out in the ISO standards provide quantitative tools to measure the biomimetic performance of the solutions developed.

### 2.2. The Principles of Living Organisms to Formalize Biomimicry

The Biomimicry Institute highlights 10 unified principles of nature (i.e., nature’s unifying patterns) or “Life’s Principle” [18], which strongly overlap those stated by Jeanine Benyus [6]:
Use materials sparingly.Use energy efficiently.Do not exhaust resources.Source or buy locally.Optimize the whole rather than maximize each component individually.Do not pollute your nest.Remain in dynamic equilibrium with the biosphere.Use waste as a resource.Diversify and cooperate.Be informed and share information.

However, it remains difficult to assess the level of implementation of some principles in a project either because they are not quantifiable or because no measurement indicators have been developed. In any case, these principles are always effective guides or sources of inspiration during the biomimetic design process. They serve as the basis for the development of biomiMETRIC assistance tool, the objective of which is to provide a quantification approach to each principle and a measurement indicator to assess biomimetic performance.

### 2.3. Biomimetic Principles from the Study of Living Organisms and Sustainable Ecosystems

By adapting the form to the function of the organism, its life is built with a minimum of resources [18]. In the context of a project or product development, a biomimetic approach invites the designer to learn about the use of resources and their environmental impacts in order not to generate the unsustainability or over-consumption of these resources. If toxic substances exist in living organisms, all toxic molecules are fairly quickly degradable. In addition, organisms only produce toxins to avoid damaging the environment and to respect its absorption capacity [18].

To evolve and adapt to their local environments, ecosystems rely on biodiversity, which is one of the keys of their resilience [19,20]. The higher the diversity, the greater the interactions and cooperation, the more effective the system will be in performing its functions despite unpredictable changes in the external environment. In a mature ecosystem, production and recycling circuits are in balance. Thus, in a forest, dead plants are recovered and degraded by organisms (e.g., fungi and bacteria). The latter redistribute the material, which is transformed in the form of nutrients, without generating waste [21]. The conditions necessary for life result from a series of exchanges between organisms and their environment. All withdrawn stocks circulate but do not decrease.

Using waste as an input for the manufacture of a product and providing other companies with its waste, which then becomes a resource, are approaches in this direction. Cooperation between companies ensures the closed circulation of resources, which is similar to that found in nature [21]. Within the framework of a biomimetic approach, it is necessary to encourage the implementation of industrial ecology or circular economy approaches by identifying opportunities for cooperation and the implementation of feedback loops.

### 2.4. Life-Cycle Assessment Method for Impact Evaluation

A wide range of LCA impact methods [22,23] is available: ReCiPe 2016, Impact 2002+, and EDP 2013 in the European context; TRACI and LUCAS in North American context; GHG Protocol, IPCC 2013, and USEtox in the global context. The main methods that are potentially relevant for the tool will be presented. However, this article does not describe in detail the methods in the literature [23]. These impact methods use characterization factors that are indicators that represent each impact category based on its own value and unit of measurement. For example, the characterization factor of radiative forcing leading to climate change is estimated in kg CO_2 eq_. Each greenhouse gas (GHG) is then converted into a kg CO_2 eq_ using a multiplicative factor.

All the methods used to assess environmental impacts have specific strengths, weaknesses, and limitations, and they all offer a tool for quantifying the impacts of midpoints and endpoints, which correspond to the aggregation of intermediate impact categories. These multiple quantitative methods are used to assess one or more different categories of impacts. It is recognized that the midpoint impact categories have fewer uncertainties than the endpoint impact categories, and they are based on more widely accepted scientific models. However, the results of these methods are often more difficult to interpret [24]. To calculate impact, each method considers a different number of chemicals in the life-cycle emission inventory (LCI).

BiomiMETRIC presented in this article prioritizes midpoint impact categories rather than the more aggregated and less precise endpoint categories. In addition, the midpoint impact categories often cover issues that are directly associated with the living world, such as ecotoxicity, biodiversity, eutrophication, and resources. The many methods have been previously analysed and compared according to the impact to be evaluated [24].

The following is a summary of the presentation of the main methods:
-Method ReCiPe, which was revised in 2016 for use in life-cycle analysis [25,26], includes the impact categories of midpoint (problem-oriented) and endpoint (damage-oriented). The characterization factors (CF) of ReCiPe are given for the following 13 impact categories [25]: (1) Climate change. CF is the global warming potential based on the IPCC 2013 report. Unit: yr/kg CO_2_ equivalents; (2) stratospheric ozone depletion. CF accounts for the destruction of the stratospheric ozone layer by anthropogenic emissions of ozone depleting substances. Unit: yr/kg CFC-11 equivalents; (3) ionizing radiation. CF accounts for the level of exposure for the global population. Unit: yr/kBq Cobalt-60 equivalents to air; (4) fine particulate matter formation. CF is the intake fraction of PM2.5. Unit: yr/kg PM2.5 equivalents; (5)a photochemical ozone formation and human health. CF is determined from the change in the intake rate of ozone due to changes in the emission of precursors (NOx and non-methane VOC). Unit: yr/kg NOx eq; (5) b photochemical ozone formation and terrestrial ecosystems. CF is determined from the change in the intake rate of ozone due to changes in the emission of precursors (NOx and non-methane VOC). Unit: yr/kg NOx eq; (6) terrestrial acidification. CF is acidification potential (AP) derived using the emission weighted world average fate factor of SO_2_. Unit: yr/kg SO_2_ equivalents; (7) freshwater eutrophication. CF accounts for the environmental persistence (fate) of the emission of P (phosphorus)-containing nutrients. Unit: yr/kg P to freshwater equivalents; (8) marine eutrophization. CF accounts for the environmental persistence (fate) of the emission of N (nitrogen) containing nutrients. Unit: yr/kg N to marine equivalents; (9) ecotoxicity terrestrial, freshwater, marine, and human carcinogenic or human non-carcinogenic. The CF of human toxicity and ecotoxicity accounts for the environmental persistence (fate) and accumulation in the human food chain (exposure), and toxicity (effect) of a chemical. Unit: yr/kg 1,4-dichlorobenzeen (1,4-DCB) emitted. (10) water use. CF is the amount of fresh water consumption. Unit: m^3^ water consumed. For the moment, this impact category does not include regionalized characterization factors. The AWARE method is better to evaluate impact on water; (11) land use. CF is the amount of land transformed or occupied for a certain time. Unit: m^2^*yr; (12) mineral resource scarcity. CF is the surplus ore potential. The unit is kg copper (Cu) equivalents; (13) fossil resource scarcity: CF is the fossil fuel potential, based on the higher heating value. Unit: kg oil equivalents. The characterization factors are representative of the global scale.-Method CED [27] enables the evaluation and comparison of energy criteria for products and services. The primary energy demand and all energy carriers that are found in nature are calculated for the entire lifetime. CED is the sum of the cumulative energy demands for the production, use, and disposal of the product. This method was published by Ecoinvent, and it is widely used as a screening indicator for environmental impacts [28]. The CF [23] of the energy resources are divided into five impact categories: (1) Non-renewable, fossil (based on the upper heating value of the resources); (2) non-renewable, nuclear (based on the upper heating value of the resources); (3) renewable, biomass (based on the amount of energy harvested or converted); (4) renewable, wind, solar, and geothermal (amount of energy harvested or converted); 5) renewable, water (based on the amount of rotation energy transmitted to the turbine). The unit of CF is megajoule equivalent (MJ eq), and the method has a global location (no regionalization).-Method CExD. Exergy is a way to express the quality of energy rather than the context of energy. Exergy is a measure of the useful work that a certain energy carrier can offer. For instance, natural gas has a high exergy value, as it can be used to create high temperatures and highly pressured steam. In this method, exergy is used as a measure of the potential loss of useful energy resources. In order to quantify the life-cycle exergy demand of a product, the indicator cumulative exergy demand (CExD) is defined as the sum of exergy of all resources that are required to provide a process or product. [26]. The CF of the exergy resources are divided into 10 impact categories: (1) Non-renewable fossil; (2) non-renewable nuclear; (3) renewable kinetic; (4) renewable solar; (5) renewable potential; (6) non-renewable primary; (7) renewable biomass; (8) renewable water; (9) non-renewable metals; (10) non-renewable minerals. The unit of CF is megajoule equivalent (MJ eq).-Method AWARE [26] consists of the following: (1) A water-use indicator representing the relative available water remaining (AWARE) per area in a watershed after the demand of humans and aquatic ecosystems has been met; (2) the recommended method from WULCA (i.e., a working group under the umbrella of the UNEP-SETAC Life Cycle Initiative) to assess water consumption impact in LCA. It is used to assess the potential of water deprivation in either humans or ecosystems (the less the available water remaining per area, the more likely it is that another user will be deprived). The AWARE indicator is calculated in two steps: (1) Relative to the area (m^3^/m^2^·month), the AWARE indicator (availability minus demand [AMD]) of humans and aquatic ecosystems; (2) the value is normalized with the world average result (AMD = 0.0136 m^3^/m^2^·month) and inverted. The result represents the relative value in comparison with the average m^3^ consumed in the world. The world average is calculated as a consumption-weighted average. The AWARE indicator is non-dimensional and ranges from 0.1 to 100, where 0.1 is the best target. A value of 1 is the world average; a value of 10 describes a region where there is 10 times less available water remaining per area than the world average.-Method IPCC 2013, which was developed by the International Panel on Climate Change (IPCC), lists the climate change factors with a timeframe between 20 and 100 years. This method is the most frequently recommended for GHG evaluation, and IPCC is used in many other methods. Direct global warming potentials (GWPs) are an index for indicating the impact of carbon dioxide and estimating the relative global warming contribution of a kg of a particular GHG compared to the emission of a kg of CO_2_ [28]. The characterization factors of IPCC, expressed in kg CO_2_ eq, are available in table based on a 100-year time horizon [29].-USEtox2 [26,30] is an environmental model for the characterization of human and ecotoxicological impacts. This method is designed to describe the fate, exposure, and effects of chemicals. It is officially recommended as an assessment method by the European Commission, the World Business Council for Sustainable Development, and by the United States EPA. The CF [31] are as follows: (1) Human toxicity potential, where CF is expressed in comparative toxic units (CTUh), providing the estimated increase in morbidity in the total human population per unit mass of a contaminant emitted and assuming equal weighting between cancer and non-cancer effects. Unit: Disease cumulative cases/kg; (2) ecotoxicity potential. The ecotoxicological effect reflects the potentially affected fraction (PAF) of species. Unit: PAF m^3^ kg^−1^. The CF is representative of the global scale.

The impact methods most commonly used in the scientific community working in the field of LCA are implemented in databases such as Ecoinvent [15], which are available in a wide range of software, such as Simapro [23] or Open LCA [32]. Free impact assessment methods that are compatible with Open LCA are available on the Nexus Open LCA website [33]. The documentation accompanying these software packages presents the methods and impact categories analyzed [34].

## 3. Methodological Approach

### 3.1. Dimensions of Biomimetic Eco-Innovation

In this article, we propose a new approach by aggregating the 10 “Life’s Principles” into three dimensions of biomimetic eco-innovation, as shown in Figure 2.

These dimensions are as follows:
Efficiency and frugality: This dimension includes the principles of “Using materials and energy sparingly and effectively,” “Source or buy locally,” “Do not exhaust resources,” and “Optimizing the whole rather than maximizing each component individually.” Efficiency is associated with frugality in resource consumption.Preservation and resilience: This dimension includes the principles of “Do not pollute your nest” and “Remain in dynamic equilibrium with the biosphere.” Preservation is associated with resilience, which ensures that nature adapts to change.Circularity and systemic approach: This dimension includes the principles of “Use waste as a resource,” “Diversify and cooperate,” and “Be informed, share information, and implement feedback loops.” Circularity is associated with the systemic approach based on the analysis of flows into and out of the system boundaries.

### 3.2. Association of Environmental Impact Assessment Methods Used in LCA with the Principles of Living Organisms which Guide the Biomimetic Design

In this approach, which is aimed at developing biomiMETRIC assistance tool, each principle of living organism is associated with an impact assessment method that is used in LCA to quantify this principle. For each “Life’s Principles” and each dimension of biomimetic eco-innovation presented in Figure 2, we will associate an impact method with a relevant indicator of the environmental aspect to be analyzed.

❖The efficiency and frugality dimension includes the five biomimicry principles described in detail below. Efficiency is associated with frugality in resource consumption. Each impact method that is proposed to analyse a principle is also presented.

Use materials sparingly. This principle is proposed to reduce the quantities of materials and ores used. For this principle, we will use the ReCiPe method [25] to evaluate the consumption-mineral resource scarcity. The other impact categories of the ReCiPe method previously presented will be used to quantify other biomimetic principles or “Life’s Principles”.Use energy efficiently. This principle is proposed to maximize the efficiency of the energy conversion systems used, especially in the case of non-renewable resources and energy of mineral resources with the impact category or characterization factor, 12y. For this principle, we use the cumulative energy demand (CED) method to estimate the amount of energy used.Do not exhaust resources. This principle invites us to focus on abundant, renewable, and easily accessible resources and to be aware of resource limitations or renewal rates. For this principle, we use the cumulative exergy demand (CExD) method to assess the quality degradation of the energy used. We will also use the ReCiPe method [25,26], which was presented above, to assess fossil resource consumption using the 13 Fossil resource scarcity impact category. For this principle, we also recommend the AWARE method for assessing the impacts on water availability [35]. Many impact analysis methods, such as ReCiPe, do not sufficiently consider the impact of water consumption on its availability and on human health [36].Source or buy locally. This principle concerns the reduction of the impact of transport, especially in terms of GHG emissions. We use the IPCC 2013 GWP 100a method to assess emissions in kg CO_2_ eq.Optimize the whole rather than maximize each component individually. The objective of this principle is to define optimization or quality according to the principles of sustainable development. No impact quantification method is applicable to this principle. To evaluate it, in biomiMETRIC, we formulate the series of five questions presented below, which are evaluated and weighted according to their level of consideration in the product or solution. The higher the score, the more strongly the question is considered from the point of view of the implementation of biomimicry in the product or the design.

The following questions are implemented in BiomiMETRIC:
Are repair, repackaging, dismantling, and recycling at the end of life possible?Is the quantity of residual materials reduced?Are the product and production aimed to respect the environmental support capacity?Do the product and production reduce environmental impacts?Is an ecolabel or environmental product declaration being sought?

❖The preservation and resilience dimension includes the principles presented below. Preservation is associated with resilience, which ensures that nature adapts to change because it is consistent with the principles:

6.Do not contaminate your nest. This principle invites us to use green chemistry with low environmental impacts and to avoid the generation of pollutants. For this principle, we use the ReCiPe method [25,26] presented above to evaluate GHG emissions with the characterization factor 1, climate change, the depletion of the ozone layer with the characterization factor 2, stratospheric ozone depletion, the assessment of fine particulate matter emissions PM 2.5, PM 10 with characterization factor 4, fine particulate matter formation, and the formation of photochemical smog with characterization factor 5, photochemical ozone formation.7.Remain in dynamic equilibrium with the biosphere. This principle invites us to minimize environmental impacts so that the capacity of the biosphere is not degraded in supporting life. For this principle, we use the ReCiPe method [25,26] presented above to evaluate the acidification of the environment with characterization factor 6, terrestrial acidification, the eutrophication of fresh water and oceans with the characterization factors 7, freshwater eutrophication and 8, marine eutrophication, human toxicity and aquatic and terrestrial ecotoxicity with characterization factor 9, ecotoxicity terrestrial, freshwater, marine, and human carcinogenic or human non-carcinogenic.

In order to standardize the characterization of the impacts of chemical compounds at the human and ecotoxicological levels, the *USEtox2* method was developed under the aegis of the UNEP/SETAC Life Cycle Initiative [36]. We will use this method to assess impacts such as soil acidification or water ecotoxicity. The *USEtox2* method is widely accepted in its field, and it is used in both Europe and North America [37] to assess the impacts of chemicals, both on humans (increase in cases of disease per kg of substances emitted) and on ecosystems (fraction of species affected per kg of substances emitted).

❖The circularity and systemic dimension includes the principles presented below. These principles are qualitative and cannot be quantified using impact methods in LCA. To overcome this problem, we will develop a series of questions and a scorecard to quantify the performance of a solution regarding the principles of biomimicry.

8.Use waste as a resource. This principle invites us to close the loop in a circular economic logic. To evaluate this principle, we use the ReCiPe method [25,26] presented above to quantify land use with the characterization factor 11, land use. A closure of flows and a reduction in the amount of waste will reduce the demand for ground space to ensure the supply of resources and waste management.9.Diversify and cooperate. This principle invites us to implement the circular economy. No impact quantification method is applicable to this principle. To evaluate it, we use in biomiMETRIC the series of five questions presented below, which are evaluated and weighted according to the level of consideration of the questions in the product or solution. The higher the score, the more strongly the question is considered from the point of view of the implementation of biomimicry in the product.

The following questions are implemented in BiomiMETRIC:
Are functional economy (or cooperation economy) tools known and used?Can the project be positioned in a non-competitive niche to avoid competition?Have stakeholders been identified and consulted during the project?Is the systemic approach used?Are circular economy principles known and applied?

10.Be informed, share information, and implement feedback loops. This principle invites us to adopt a systemic vision. Circularity is associated with the systemic approach based on the analysis of flows into and out of the system boundaries. Living organisms receive and process a great amount of information to adapt to variations in their environment (e.g., temperature, season, and light). For this principle, we also formulate a series of five questions presented below.

The following questions are implemented in BiomiMETRIC:
Is the information required to increase sustainability disseminated and accessible?Are training and evaluation on the biomimetic approach proposed?Does the company integrate social and environmental information in its decision-making?Is a special attention paid to scarce and over-exploited resources?Are the major issues related to sustainability and societal inequalities known and shared?

We present a summary of the association of each biomimetic “Life’s Principles” with the methods from life-cycle analysis in Table 1.

### 3.3. Development of BiomiMETRIC Assistance Tool for Biomimetic Design

This article provides the foundations and characteristics of BiomiMETRIC, a tool in the form of spreadsheets to assist in biomimetic design. BiomiMETRIC will help to quantify the principles and strategies of living organisms to assist the engineer, architect, or designer in the biomimetic design process. Admittedly, not all living principles and strategies could always be implemented in the process of biomimetic eco-innovation. However, by considering as many biomimetic criteria as possible, the solution may be developed for greater consistency with sustainability issues.

The “Life’s Principles” have been grouped according to their similarity, and a dimension of biomimetic eco-innovation is proposed for each group. To evaluate the biomimetic performance of a product, project, or concept, BiomiMETRIC is aimed to quantify each principle of living organisms using the characterization factors and indicators derived from the environmental impact assessment methods recognized in the life-cycle analysis. The principles that cannot be quantified by impact methods will be assessed using weighted questions. The treatment of each question will be carried out as follows:
Weight the question according to its importance in the context of the project. The weighting is on a scale of 1 to 5, and it serves as a calibration of the weight that is given to the question in the context. A weighting of 1 means that it is desirable for the question to be considered, and a weighting of 3 indicates that the question is necessary, whereas a level 5 indicates the indispensable need to consider the question.Evaluating the question is equivalent to assigning a score that is (−−), (−), 0, (+) or (++), depending on how the project, concept, or product answers the question.Subsequently, according to the weighting and evaluation of the question, a score between −2 and 2 is granted according to the values presented in the matrix shown in Table 2.

We performed a distribution of scores as shown in Table 2 so that a high weight (5) gives a high score (−2 or +2) based on a significant evaluation (− −) or (++). The distribution is not linear in order to reinforce the impact (negative or positive) related to strong evaluations (− −) or (++) and to give less importance to the more neutral evaluations (−; 0; +). We also chose to rate the positive aspects more strongly than highlight the negative aspects. For example, for the same weighting of a question equal to 4, if the question is evaluated (−), it will receive a score of (−1) whereas if it is evaluated (+) it will receive a score of (+1.35).

For each principle, the average of the scores obtained for each question is then calculated. The average value of the notes to the question series is represented on a graph under the “qualitative results” tab in the biomiMETRIC tool. This window will be presented later as part of the analysis of an example.

## 4. Results

BiomiMETRIC assistance tool, presented in this article synthesizes, presents, and interprets all the results of the LCA methods used to evaluate the biomimetic principles (“Life’s Principles”). It also presents the answers to questions about the biomimetic approach used in the design. Although the tool presents the results to quantify certain impacts of a biomimetic design perspective on a given solution, it is recommended that a comparison be made between several solutions to determine which has the best biomimetic performance. The assessment is therefore relative rather than absolute because we do not yet have thresholds that can indicate, for example, the maximum acceptable greenhouse gas level for considering that a solution is biomimetic.

### 4.1. Procedure for Using BiomiMETRIC

The steps in using the tool to conduct the analysis are presented in Figure 3 and described below.
Using LCA software such as Open LCA [32] or Simapro [23], which are equipped with databases such as Ecoinvent [16,28], select the materials and processes that constitute Scenario A, the biomimetic performance of which you wish to evaluate.Select the impact assessment methods recommended in the tool for each biomimetic principle. The methods used are those presented: ReCiPe, USEtox2, IPCC 2013, AWARE, CED, and CExD. Evaluate each impact category and record the results associated with each biomimetic principle presented in the tool (one principle per tab in the spreadsheet).For each biomimetic principle, in addition to the results of the LCA methods, evaluate the questions asked using the evaluation and weighting grid available in BiomiMETRIC. A quantitative value will then be calculated by BiomiMETRIC based on the level of integration of the questions asked in the project, the concept, the product, or the design.Repeat the sequence for scenario B, and compare its performance with scenario A.After recording the results of the LCA for the two scenarios A and B to be compared, evaluate the questions on the 10 “Life’s Principles:
(a)Open the *quantitative results* Table. This sheet will present a summary of the results and the scenario that best meets the biomimetic principles.(b)Open the *quantitative results graphs* tab, where the compared results of the two scenarios are presented in graphical form.(c)Open the *qualitative results* Table. This sheet will present a performance evaluation based on the answers to the qualitative questions about the biomimetic approach implemented in the project. In this section, there is no comparison between the scenarios A and B, as we consider that the general principles evaluated by the questions will be implemented in a similar way regardless of the scenario chosen. The sheet also presents the biomimetic principles for which corrective action is required.

### 4.2. Using BiomiMETRIC: An Illustration

BiomiMETRIC quantitative performance tool provides a comparison of the biomimetic performance of scenarios A and B. To demonstrate the potential use of the tool, we compared two insulation materials: Stone wool and cork panels. The characteristics of these two alternatives, which can be used in a biomimetic architectural project, are presented in Table 3. The impacts of these materials are considered over the entire life cycle in this analysis (i.e., extraction, processing, implementation, and end-of-life management are taken into account).

The same thermal insulation will be achieved by using 70 kg of stone wool or 110 kg of cork panel for the walls in an architectural project, which we will assume is designed following the biomimetic approach described in ISO 18458. Both materials are evaluated using the 10 principles of biomimetic design integrated into BiomiMETRIC, as well as using the qualitative questions to reveal which of the two insulators has the best biomimetic performance. BiomiMETRIC allows us to see that in terms of material consumption, for example, stone wool has a greater impact than cork. It appears that the impacts on resource use (ReCiPe 2016/mineral resources scarcity) are 0.391 and 0.287 kg copper equivalent, respectively, for stone wool and cork. Stone wool, therefore, puts more pressure on resources than cork does. The indicator used for resource scarcity assessment indicates the amount of ore to be extracted to obtain 1 kg of copper, which is considered a reference resource. The scarcer the resource becomes, the greater the necessity to dig the earth and extract ore to obtain 1 kg of copper. Each material is converted into kg of equivalent copper via a conversion factor presented in the method documentation [25]. A material therefore has the higher potential for scarcity if its numerical value in kg Cu eq is high.

The quantitative results dashboard of BiomiMETRIC presented in Figure 4 shows the impact of each scenario (A or B) in relation to the sum of the impacts of the scenarios (A + B). For example, in graph 1 of the dashboard, entitled resource consumption, cork (in red) represents 42.3%, and stone wool (in blue) represents 57.7% of the sum of the total impacts expressed in kg Cu eq in this category. Stone wool therefore has more impacts than cork has.

The analysis of the graphs of each quantifiable principle shows that stone wool is relatively more efficient in terms of biomimicry because it has lower environmental impacts in terms of energy demand, degradation of energy quality (exergy), consumption of fossil resources, water consumption, GHG emissions, accumulation of toxic substances, acidification of water, eutrophication, and land use. In this category of quantitative results, stone wool has superior biomimetic performance compared to cork.

The principles that cannot be quantified by LCA methods are assessed based on the answers to the questions, as explained previously. The results show the overall strengths and weaknesses of the project, product, or concept. This analysis does not present a comparison of scenarios A and B; instead, it shows the overall assessment of the implementation of the biomimicry principles in the project. Figure 5 presents a summary of the qualitative results for each principle. The figure shows that the project using the insulating materials (stone wool or cork) warrants corrective action in terms of integrating principle 8, Use waste as a resource (score = 0.25 out of 2), which could be implemented using a circular economy approach. Principle 9, *diversification and cooperation*, also needs to be improved (score = 0.5 out of 2) by using, for example, the product-service systems (PSS) [40] approach, which would offer the customer insulation performance and guaranteed comfort in addition to the insulation material.

## 5. Discussion

To facilitate the design of sustainable solutions, engineers and designers need tools to guide them in their choices. These tools must integrate not only the main principles proposed in the biomimetic design approach but also the quantitative approaches to impact assessment, such as those proposed by the life-cycle analysis. The main results of this research demonstrate that BiomiMETRIC, the tool developed and presented in this article facilitates decision-making in choosing the scenario that is the most in line with the biomimetic principles. Thus, in the very brief example in which we compared two insulating materials, the results showed that stone wool offered a better biomimetic performance compared with cork, which is a bio-based material. It is mainly in terms of water consumption, GHG emissions, energy demand, energy quality degradation, and land use that stone wool is more advantageous than cork, and it is more efficient in terms of low material consumption.

BiomiMETRIC has therefore made it possible to quantify the biomimetic principles that have so far been evaluated subjectively. The use of the tool validates the fundamental principles of biomimetic design as presented in the ISO 18458 standard or by the Biomimicry Institute using methods derived from life-cycle analysis. In conducting a conventional life-cycle analysis, not all the methods selected for each “Life’s Principles” analyzed in BiomiMETRIC are used. Moreover, a conventional LCA that is carried out in accordance with the ISO 14040 and 14044 Standards [41] requires specialized expertise, which may not be readily available in all design teams.

The answers to the qualitative questions revealed that regardless of the material, the design approach had weaknesses in terms of the principle of using waste as a resource (industrial ecology) and the principle of diversification and cooperation (product-service systems). These principles could be improved by taking into account the questions asked and the answers. Initially and before the analysis, it could have been obvious that a bio-based material such as cork has a higher biomimetic performance than an industrial material such as stone wool, and yet the analysis showed the opposite.

## 6. Conclusions

In this article, we proposed a functional tool to achieve the linkage between LCA and biomimetic principles: BiomiMETRIC assistance tool. This tool facilitates quantification and decision-making in the biomimetic approach to help to design sustainable products or sustainable projects. We also proposed a group of biomimetic principles according to three dimensions: (1) Efficiency and frugality; (2) preservation and resilience; (3) circularity and a systemic approach. BiomiMETRIC allowed us to compare two scenarios and determine the one with the best biomimetic performance according to the principles proposed by the Biomimicry Institute, which are evident in sustainable ecosystems. BiomiMETRIC could be very useful for designers and engineers that seek design solutions using a biomimetic approach.

It should be recalled that a significant part of the work in developing the tool presented in this article was aimed to analyze and justify the choice of the method resulting from the life-cycle analysis, which was the most appropriate for quantifying each “Life’s Principles” used in the biomimetic approach. BiomiMETRIC could be used as part of a biomimetic design approach or as a method for analyzing environmental performance as part of a conventional design process.

The newly developed impact world method [36], which is not available at the moment in the LCA software, must be tested, as it could provide a new alternative method for quantifying the principles used in the biomimetic design process. Although it is recognized and potentially adapted to the quantification of biomimicry principles, the tool for the reduction and assessment of chemical and other environmental impacts (TRACI) was developed by the U.S. EPA specifically for use in the US [26,42] was not used to develop BiomiMETRIC presented in this article. TRACI is a North American reference method used for environmental product declaration. It generally presents impact categories similar to ReCiPe or USEtox, which would lead to redundancy in the impact assessment; however, TRACI evaluates fine particulate matter PM 2.5, which is not found in other methods, which instead treat PM 10. TRACI is a midpoint-oriented life-cycle impact assessment methodology with 10 characterization factors. This method could be considered for use in the development of a second version of BiomiMETRIC.

In a subsequent work, it would be interesting to develop a biomimetic index that could be obtained by normalizing and aggregating the results of each of the biomimetic design principles presented here. The index would be between 0 and 1, which would make it easier to choose the scenario with the highest biomimetic performance.

Although the biomimetic principles can be queried, they remain a source of inspiration for achieving sustainability in processes, products, projects, and design. The consideration of changing environmental conditions and new constraints in the design of solutions is characteristic of the biomimetic approach, which is inspired by the principles of resilience and adaptation. Unlimited growth is not a sustainable or widespread behaviour in nature, where living organisms must deal with the scarcity of energy and resources. Zero waste, zero emissions, and the circularity of flows are basic principles of life in nature (“Life’s Principles”), which must be prioritized and used to guide all ecodesign approaches.

## Figures and Tables

**Figure 1 biomimetics-04-00049-f001:**
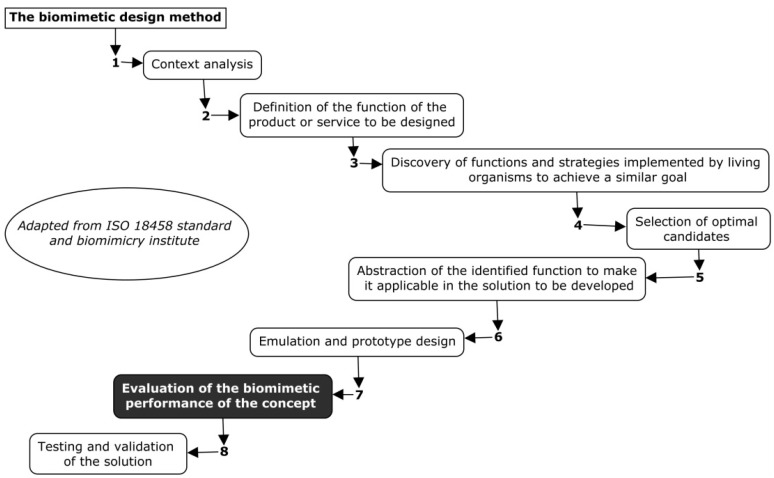
The biomimetic design method. Adapted from [3].

**Figure 2 biomimetics-04-00049-f002:**
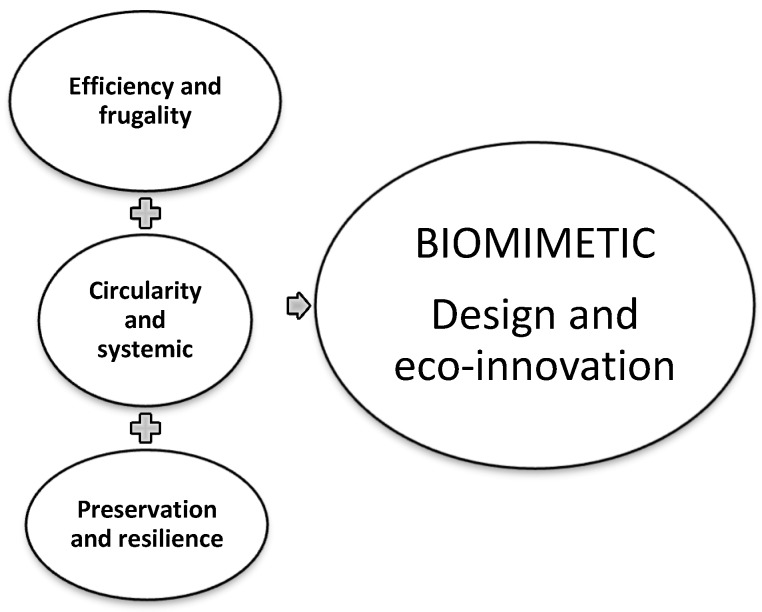
Dimensions proposed for grouping the principles of nature (“Life’s Principles”) used in biomimetic design.

**Figure 3 biomimetics-04-00049-f003:**
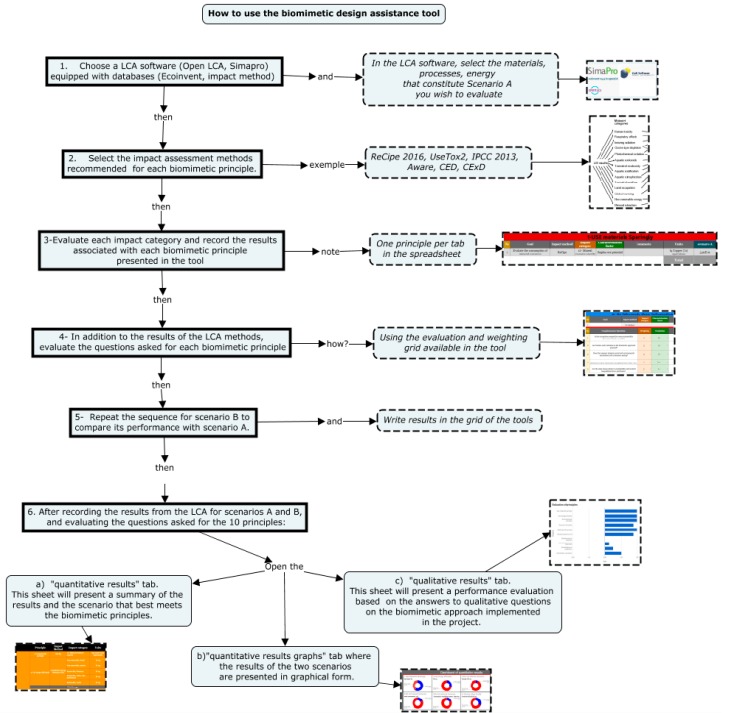
Algorithm for using BiomiMETRIC.

**Figure 4 biomimetics-04-00049-f004:**
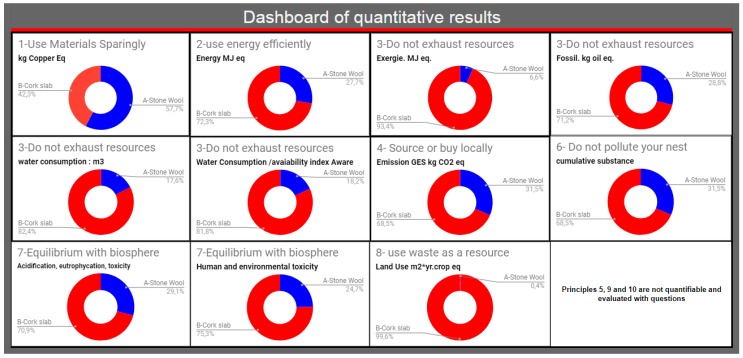
Dashboard of the quantitative results given by BiomiMETRIC for each quantifiable biomimetic principle.

**Figure 5 biomimetics-04-00049-f005:**
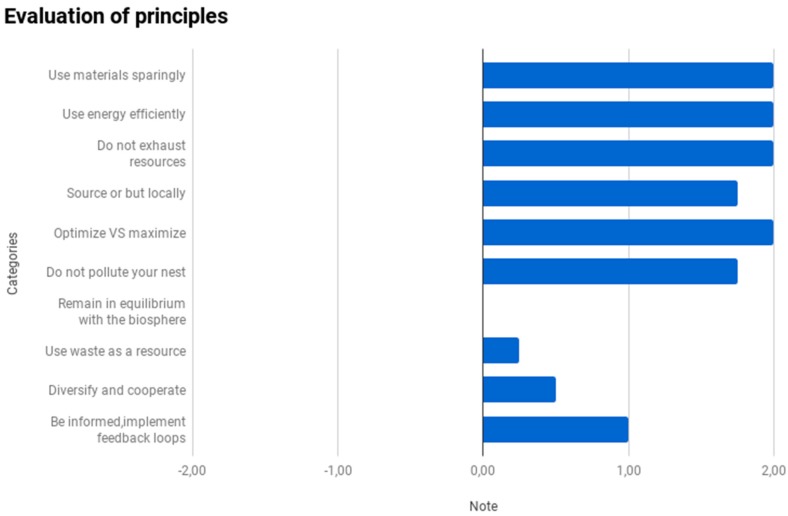
Average value of the notes to questions for each principle. Dashboard of the qualitative results given by BiomiMETRIC.

**Table 1 biomimetics-04-00049-t001:** Main impact assessment methods used for biomiMETRIC assistance tool and their association with the “Life’s Principles”.

Biomimetic Dimensions and Principles		LCA Approach			
		Method	Impact Category	Information	Unit
**Efficiency and frugality**					
1	Use materials sparingly	ReCIPe	12 - Mineral resource scarcity	Consumption of mineral resources	kg Cu eq
2	Use energy efficiently	Cumulative Energy Demand	Amount of energy used	Energy consumption	MJ eq
3	Do not exhaust resources	Cumulative Exergy Demand	Total exergy removal from nature. Quality degradation of the energy used	Sum of exergy of all resources required to provide a process or product.	MJ eq
		ReCIPe	13 - Fossil resource scarcity	Fossil resource consumption	kg oil eq
		AWARE	Relative Available WAter REmaining per area in a watershed after the demand (humans & ecosystems) has been met	Impacts on water availability	Index: range [0.1;100]
4	Source or buy localy	IPCC 2013 GWP 100a	GHG emissions	Global warming potential of air emissions	kg CO_2_ eq
5	Optimize the whole rather than maximize each component individually	Specific questions	5 questions evaluated and weighted according to their level of consideration in the product	Optimization or quality according to the principles of sustainable development	According to the weighting and evaluation of the question, a score between −2 and 2 is granted
**Preservation and resilience**					
6	Do not contaminate your nest	ReCIPe	1 - Climate changes	Global warming potential of pollutants	kg CO2 eq
			2 - Stratospheric ozone depletion	Destruction of the stratospheric ozone layer	kg CFC-11 eq
			4 - Fine particulate matter formation	PM 2.5 in air emissions	kg PM2.5 eq
			5 - Photochemical ozone formation	Change in intake rate of ozone due to change in emission of precursors (NOx and NMVOC)	kg NOx eq
7	Remain in dynamic equilibrium with the biosphere	ReCIPe	6 - Terrestrial acidification	Acidification Potential	kg SO2 eq
			7 - Freshwater eutrophication	Emission of P(phosphore)	kg P to freshwater
			8 - Marine eutrophication	Emission of N (nitrogen)	kg N to marine water
			9 - Ecotoxicity Terrestrial, Freshwater, Marine and Human carcinogenic or Human non-carcinogenic	Ecotoxicity accounts for the environmental persistence and accumulation in the human food chain	kg 1,4-dichlorobenzeen (1,4-DCB)
		USEtox2	Human toxicity potential	Estimated increase in morbidity in the total human population per unit mass of a contaminant emitted	Disease cumulative cases / kg substance
			Ecotoxicity potential	Potentially Affected Fraction (PAF) of species	PAF m^3^ kg^−1^
**Circularity and systemic**					
8	Use waste as a resource	ReCIPe	11 - Land Use	Amount of land transformed or occupied for a certain time	m2*yr. crop eq
9	Diversify and cooperate	Specific questions	5 questions evaluated and weighted according to their level of consideration in the product	Implement the circular economy	According to the weighting and evaluation of the question, a score between −2 and 2 is granted
10	Be informed, share information and implement feedback loops	Specific questions	5 questions evaluated and weighted according to their level of consideration in the product	Adopt a systemic vision	According to the weighting and evaluation of the question, a score between −2 and 2 is granted

**Table 2 biomimetics-04-00049-t002:** Results matrix according to the evaluation and weighting of the questions in BiomiMETRIC.

		Evaluation					
		S/O	(−−)	(−)	(0)	(+)	(++)
weighting	S/O	0	0	0	0	0	0
	1	0	0	0	0	0.2	0.4
	2	0	−0.25	−0.1	0.25	0.6	0.8
	3	0	−0.5	−0.3	−0.1	0.9	1.2
	4	0	−1.5	−1	−0.5	1.35	1.6
	5	0	−2	−1.5	−1	1.7	2

**Table 3 biomimetics-04-00049-t003:** Comparison of materials for a biomimetic architectural project to obtain the same thermal insulation performance R.

Material	Coefficient of Thermal Resistance R (m^2^ K/W) for 100 mm of Thickness	Density (kg/m^3^)	Mass Required to Insulate a 1 m^3^ Space
Stone wool (depending on quality) [38]	2.7	70	70
Cork panel (depending on quality) [39]	2.7	110	110

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
