# Peer review of "BiomiMETRIC Assistance Tool: A Quantitative Performance Tool for Biomimetic Design"

_biomimetics, 2019, doi:10.3390/biomimetics4030049_

Round 1
Reviewer 1 Report
The provide a well-structured and technically sound manuscript. The topic is of current interest and the work developed is relevant for the scientific community. No errors were found nor any corrections are required. In my opinion, the manuscript is ready for publication.
Author Response
Thank you for the positive comments and the opinion considering that the article is ready for publication (the manuscript is ready for publication).
Best Regards,
Philippe Terrier
Reviewer 2 Report
In the paper "optimisation procedure" is poorly described.
From mathematical point of view first of all the aim function has to be defined with some limitations (boundary condititions).
In the present form it looks and sounds much more as a form of sensitivity or questionary study instead to be real optimistaion procedure.
Additionally, optimisation means in this case that we want to maximse/minimise something ??? No criteria at all.
There is no justifications and discussion of values put into Table 2 to the evaluation and weighting process.
